# Rheological Properties and Early-Age Microstructure of Cement Pastes with Limestone Powder, Redispersible Polymer Powder and Cellulose Ether

**DOI:** 10.3390/ma15093159

**Published:** 2022-04-27

**Authors:** Kaiwen Feng, Zhanjun Xu, Weizheng Zhang, Kunlin Ma, Jingtao Shen, Mingwen Hu

**Affiliations:** 1School of Civil Engineering, Central South University, Changsha 410075, China; feng_kaiwen@163.com; 2Hunan Zhongda Design Institution Co., Ltd., Changsha 410075, China; mark-mkl@163.com (Z.X.); linquanzhang@yeah.net (W.Z.); 3China Railway Urban Construction Group Co., Ltd., Changsha 410208, China; shenjintao@crucg.com (J.S.); humingwen.cj@crcc.cn (M.H.)

**Keywords:** rheology, microstructure, cement-based material, organic and inorganic thickening agent, complementary effect

## Abstract

In order to study the synergistic effects of organic and inorganic thickening agents on the rheological properties of cement paste, the rheological parameters, thixotropy cement-paste containing limestone powder (LP), re-dispersible polymer powder (RPP), and hydroxypropyl methylcellulose ether (HPMC) were investigated using the Anton Paar MCR 102 rheometer at different resting times. The early-age hydration process, hydration products, and microstructure were also analyzed with scanning electron microscopy (SEM) and thermogravimetry analyses (TGA). The results showed that the addition of LP, RPP, and HPMC affected the rheological properties of cement paste, but the thickening mechanism between organic and inorganic thickening agents was different. The small amount of LP increased the plastic viscosity but decreased the yield stress of cement paste due to its dense filling effect. Adding 1% of RPP improved the thixotropic property of cement paste by 50%; prolonging the standing time could improve the thixotropic performance by as much as two times. Only 0.035% HPMC added to the cement paste increased the plastic viscosity by 20%, while the yield stress increased nearly twice. The more HPMC added, the more significant effect it showed. Cement paste compounds with LP, RPP, and HPMC balanced the yield stress and plastic viscosity and improved the thixotropy. The C-L6-R1.0-H0.035 paste presented as a pseudoplastic, its rheological indexes were close to one, and it was hardly affected by the resting time. The composite superposition effect of organic and inorganic thickening agents reduced the impact of resting time for all pastes. As the organic thickening component inhibited the hydration more than the LP promoted the hydration of the cement paste, indicating that the C-L6-R1.0-H0.035 paste remained in the particle fusion stage after curing for three days, as shown by the SEM images.

## 1. Introduction

The workability of fresh concrete has an important influence on construction quality. High fluidity, large slump, and high cohesion are important workability characteristics of modern concrete. High fluidity of cement-based materials has been widely used in engineering applications such as 3D-printing cement, grouting materials, and self-compacting concrete [1,2].

In practical engineering, cement-based materials should have good fluidity, high stability, segregation resistance, and no bleeding. Many studies have shown that the workability and mechanical properties, including the rheological properties of cementitious material, changed with time [3,4]. Therefore, methods such as adding superplasticizer or thickening agents and optimizing the mixing ratio are usually adopted to improve the yield stress and plastic viscosity of the paste so as to increase the anti-separation quality of each component in order to balance the performance of cement-based materials, including high fluidity, anti-segregation, high stability, passing quality, and filling quality through gaps [5].

In order to optimize the properties of fresh cement paste, researchers [6,7,8] had mixed mineral admixtures and chemical admixtures into cement-based materials to study their working properties, mechanical properties, and rheological properties. Though the physical properties and chemical compositions of thickening agents are different, thickening agents have a great influence on the properties of cement-based materials, especially when it is fresh. The physical properties of limestone powder, such as its specific surface area [9,10], have a significant effect on the properties of cement pastes. The nucleation rate of the hydration products of the paste decreases with the increase in limestone powder content and increases with the increased fineness of limestone. Commonly used chemical thickening agents such as synthetic viscosity-modifying admixtures and those of microbial origin mainly, including starch ether, cellulose ether, polyacrylamide, etc., have been confirmed to be highly effective for improving the rheology and stability characteristics of cement-based materials. The free water between cement mixtures is prevented from moving freely through the combination and substitution of groups on the molecular chain or the interaction between hydrogen molecules in the chemical mixture, thereby improving the viscosity of the paste [11,12]. Related research shows that [13,14,15] the incorporation of HPMC into cement-based mortar can significantly improve the water retention rate, thus reducing the available water in a cementitious mixture [16]. After adding latex powder into cellulose ether-modified cement paste, the rheological curve conforms to the variation law of the Herschel–Bulkley (H–B) model, and there is a positive correlation between the yield stress and thixotropic area. The viscosity of the cellulose ether solution and the cement paste has a compound superposition effect. The bridging and filling effect of an appropriate amount of re-dispersible polymer powder in fresh cement mortar can effectively improve the workability of the mortar and enhance the mechanical properties. The addition of a small amount of organic thickening agent can significantly change the working properties of the paste without changing its composition, which is beneficial for optimizing the rheological properties of special cement grouting.

The above studies have shown that thickening agents can effectively improve the workability of freshly mixed cementitious materials in spite of the small dosage. Currently, the majority of studies only show the effect of organic or inorganic thickening agents on the working properties of cement pastes, and less literature is available on the rheological properties of composite thickening agents incorporated into cementitious materials. It can be concluded from existing literature that inorganic and organic thickening agents take on different thickening mechanisms on cementitious materials. The combination of different thickening agents can play a significant role in the synergistic effect and further improve the stability of fresh cementitious materials, which is also an important direction for enhancing thickening agents for cement-based materials. However, there are few studies on how composite thickening agents improve the properties of cement-based materials and explain this from a rheological perspective. The research in this field should be further developed.

Very fine limestone powder (LP) has a thickening effect, and a replacement for cement particles is conducive to sustainable environmental development. The organic thickening agents of re-dispersible polymer powder (RPP) have a film-forming effect, and hydroxypropyl methylcellulose ether (HPMC) has a significant effect on concrete thickening and prevents bleeding. Both RPP and HPMC have the effect of improving concrete workability. In this paper, the thickening agents, LP, RPP, and HPMC, were chosen and added to cement pastes with a range of contents to study the rheological properties at different resting times. Additionally, the early-age hydration products and microstructure of the cement paste were studied to further reveal the rheological properties, thereby providing a theoretical basis for the preparation of highly stable cementitious materials.

## 2. Materials and Methods

### 2.1. Raw Materials and Admixture Proportion

P.I 42.5 Portland cement (C), which satisfied Chinese Standard GB 175, was provided by the China Building Materials Research Institute Co. Ltd. The chemical and mineral components of Portland cement are shown in Table 1 and Table 2, and its physical properties are shown in Table 3. Qualified and densified limestone powder (LP) had a specific area of 1104 m^2^/kg, and the $D\left(50\right)$ was 7.077 μm, and the CaCO_3_ content in the LP was more than 99.9%. The Redispersible Polymer Powder (RPP) is a white powder with a solid content of 99.0% and a PH value of 6–8, and it is a commercially available vinyl acetate and ethylene copolymer rubber powder. Hydroxypropyl methylcellulose ether (HPMC) uses cellulose ether with a viscosity of ${1\;\times \;10}^{5}\;\mathrm{mPa}\text{·}s$ and a molecular weight of 100,000~150,000. The number of hydroxyl groups replaced by the reaction reagent, which is also called the degree of substitution $\left(DS\right)$ of HPMC, is about 1.3~2.1. Moreover, the average amount of substituted ether groups bound on each dehydration unit, also called the molar degree of substitution $\left(MS\right)$, is about 0.1~1.0. Generally speaking, the DS of cellulose ether is the sum of the two and MSDS indicates the length of the side chain [17]. Polycarboxylic acid superplasticizer (SP) with a water-reducing rate of 32% and solid content of 33.1% was used to enhance the flowability of the fresh cement-based material. The experiment used tap water as the mixing water (W).

The particle size distribution of cement and limestone powder was measured by a Topsizer laser particle size analyzer manufactured by Zhuhai, China Omec Instrument Co., Ltd., with a dry measuring method. The particle size distribution of C and LP is exhibited in Figure 1; the medium particle size and specific surface area are shown in Table 4.

The mixture proportions are listed in Table 5. Twelve groups of mixtures were prepared. Each mixture was maintained at the water to binder ratio (W/B) 0.32 and superplasticizer (SP) 0.17%. LP replaced cement with equal mass, and the replacement amount was 3% or 6%, which was denoted as LP3 and LP6. RPP was mixed at 0.5% and 1.0% and was denoted as R0.5 and R1.0. HPMC was mixed at 0.035% and 0.075% and was denoted as H0.035 and H0.075. Sample C served as the control group, which was a blank cement paste.

### 2.2. Testing Methods

#### 2.2.1. Rheological Test and Analysis Model

An Anton Paar MCR 102 rheometer produced by the Anton Paar Company in Graz, Austria (seen in Figure 2), was used to determine the rheological curves of different pastes, with coaxial blade rotor was used for the rheology measurement. In order to ensure that the materials mixed uniformly, an electric one-phase mixer was used to mix the pastes. Firstly, water and superplasticizer were mixed together by the electric mixer at 62 rpm for 1 min, then the cementitious materials were added, and the electric mixer stirred the paste. In the process of stirring the paste, it was first stirred for 60 s at a low rate of 62 rpm, then stopped for 10 s, and then quickly stirred at 125 rpm over a time span of 60 s. Before the rheology test, the prepared fresh composite paste was allowed to rest for 15 min and 90 min, respectively.

The rheological test was conducted as follows: firstly, the shear rate increases logarithmically from 0.1 s^−1^ to 300 s^−1^ within 270 s, then kept at 300 s^−1^ to shear the paste for 120 s, followed by the shear rate decreases logarithmically to 0.1 s^−1^ within 270 s.

The rheological properties of cementitious materials are generally simplified by a specific rheological model because of their complexity. In this paper, the Bingham model, a linear formula, was used to fit the plastic viscosity of the paste, and the exponential model named Herschel–Bulkley (H–B) model was used to describe the rheological index and yield stress. This is because the rheological behavior index fitted the H–B model can better describe the degree of shear thinning or shear thickening of paste. Additionally, because the yield stress of fresh cement-based paste is prone to a small value, fitting the yield stress with the linear formula could lead to negative yield stress, which is inconsistent with reality. Therefore, fitting the yield stress with an exponential model could yield more accurate results. The rheological property parameters are fitted and analyzed by two macroscopic mechanical models: the Bingham model and H–B model, seen in Equations (1) and (2), respectively [18].
(1)$$\tau =\tau_{0}+\eta \gamma$$
(2)$$\tau =\tau_{0}+K\gamma^{n}$$
where *τ* (Pa) is the shear stress, *τ*_0_ (Pa) is the yield stress, *γ* (s^−1^) is the shear rate, *η* (Pa·s) and *K* ($\mathrm{Pa}\cdot s^{n}$) are plastic viscosity coefficients fitted by the Bingham model and H–B model, respectively. The intensity of shear thickening or shear thinning of fresh cement paste can be characterized by rheological index *n* in the H–B model. It is generally believed that shear thickening occurs when *n* > 1, and shear thinning occurs when *n* < 1. The greater difference between rheology index (*n*) and 1, the stronger the intensity of shear thickening/thinning is [19].

During the shear stress testing, the area of the hysteresis loop formed by shear stress in the ascending and descending stage can show the thixotropy of the paste and can also be called the thixotropy area (seen in Figure 3).

#### 2.2.2. TG-DTA Test

The STARe SW thermal analyzer was used to carry out the experiment. Samples cured for 3 days were put into the crucible in the furnace body, then the temperature was raised from 30 °C to 1050 °C at a heating rate of 10 °C/min. The working protective gas was nitrogen. The thermogravimetric curve (TG) and differential curve (DTA) were obtained due to the physical and chemical changes of the samples during the heating process. It can be found that a peak appearing on the DTA curve corresponds to the mass loss on the TG curve, and the product content can be determined.

#### 2.2.3. SEM Analysis

The FEI Quanta FEG 250 environmental field scanning electron microscope (SEM) was used to observe the micro-morphology of each paste cured for 3 days. The fully dried samples were placed in a high vacuum environment to observe the microstructure of the materials. The working distance was 10 mm, the acceleration voltage was 30 kV, and the detector type was SE.

## 3. Results and Discussion

### 3.1. Rheology of C-LP Paste

#### 3.1.1. Rheological Curve

Figure 4 shows the influence of the LP content on the rheological curves of pastes under different resting times. Group C represents the blank cement paste, and it is a control group. It can be seen from Figure 4 that the LP content and the resting time have a certain influence on the rheological properties of the cement pastes.

As can be seen from Figure 4a, with the increase in shear rate, the shear stress and its growth rate for all groups increase, but in the case of more LP content, lower shear stress shows. The growth rate of shear stress in C-LP pastes is higher than that of blank cement paste in the later stage. After resting for 90 min, when the shear rate is higher than about 230/s^−1^, C-LP pastes show higher shear stress than that of the cement paste. It may be due to the significant fineness of LP, which is embedded into the cement particles to produce a thickening effect. When the shear rate is high, the LP with less content is embedded to a greater extent and plays an obvious thickening effect, resulting in the increase in shear stress.

As can be seen from Figure 4b, when the shear rate increases from 0 to 50 s^−1^, the apparent viscosity of each paste decreases rapidly, indicating the pastes have yielded. When the shear rate rises greater than 50 s^−1^, the apparent viscosity of blank cement paste decreases slowly with the increase in shear rate, but the apparent viscosity of the C-LP pastes increases to a certain extent. The blank cement paste shows a slight thinning of shearing, while the C-LP pastes show a slight shear thickening. The greater the content of LP, the lower the apparent viscosity of the paste when the shear rate is higher than 100 s^−1^. Therefore, even a small amount of incorporated LP changes the rheological curve of cement paste at a high shear rate and improves the stability of the paste.

By increasing the resting time, the rheological property of the paste remains unchanged, but the shear stress and apparent viscosity increase. However, after resting for 90 min, the shear stress of C-L6 paste is lower than that of blank cement during the whole process of the rheological test.

Table 6 presents the rheological parameters of C-LP pastes. It can be seen from Table 6 that the rheological index goes up with the addition of LP after resting for 15 min. After prolonging the resting time, the C-L3 paste still takes on dilatant (*n* > 1), while C-L6 paste changes into pseudoplastic (*n* < 1). From the point of view of actual construction, the degree of dilatant or pseudoplastic is determined by whether the paste can be successfully constructed in the pumping process. In 3D printing engineering, shotcrete, and pumping construction, it is hoped that concrete can have the characteristics of shear thinning so that it can avoid negative phenomena such as sticking to pipes and interrupting the construction process during high-speed pumping.

#### 3.1.2. Yield Stress and Plastic Viscosity

Figure 5 shows the influence of the LP content and resting time on yield stress $\left(\tau_{0}\right)$ and plastic viscosity.

As can be seen from Figure 5, $\tau_{0}$ reduces with the addition of LP. After prolonging the resting time to 90 min, the $\tau_{0}$ of each paste increases but is still lower than the blank cement paste. In the C-LP pastes, the yield stress is mainly affected by the interaction force between the particles, which is determined by the particles’ surface spacing. J. Xiao et al. found that when the total specific surface area of C-LP particles increased in the range of 347~390 $m^{2}\cdot {\mathrm{kg}}^{-1}$, the particle surface spacing tended to increase, causing the total inter-particle interaction force such as van der Waals force and acid-base force between particles in the paste exhibited as the repulsive force, eventually leading to a decrease in yield stress. Additionally, the interaction between cement particles is about ten times stronger than that of the interaction between C-LP particles or LP-LP particles, causing the yield stress of pastes mixed with LP to be less than that of pure cement paste [20]. In this paper, when the content of LP in the C-LP paste increased from 0 to 6%, the total specific area of particles in the paste increased from 346 $m^{2}\cdot {\mathrm{kg}}^{-1}$ to 349.5 $m^{2}\cdot {\mathrm{kg}}^{-1}$. Thus, the $\tau_{0}$ in C-LP pastes decreased with the increasing content of LP.

It can be seen from Figure 5 that a small amount of LP increases the plastic viscosity of the pastes. However, when the LP content increased to 6%, the plastic viscosity decreased but still increased slightly compared with that of pure cement paste. When the resting time was extended, the plastic viscosity of each paste increased. During the dynamic shearing process, LP plays a major role, including in the filling and diluting effect in C-LP pastes, and the crystalline nucleation effect of LP in the hydration process with a lower content has a greater influence on the formation of the plastic viscosity. In addition, the finer LP optimizes the particle size distribution, improves the flocculation structure, and makes C-LP pastes accumulate and compact easily [21]. In summary, the addition of a small amount of finer LP into cement paste will increase the plastic viscosity.

#### 3.1.3. Thixotropy

Figure 6 shows the thixotropic area of C-LP pastes at different resting times. It is clear that the thixotropic area of C-LP paste decreases gradually with the increasing content of LP at the same resting time. Moreover, the thixotropic area of each paste goes down with the increased resting time.

According to the above results, it can be concluded that the addition of a small amount of LP changes the paste from a shear-thinning state to shear-thickening; that is, the paste changes from a pseudoplastic to a dilatant after adding RPP. Furthermore, the addition of LP decreases the yield stress and thixotropy but increases plastic viscosity.

### 3.2. Rheology of C-RPP Paste

#### 3.2.1. Rheological Curve

Figure 7 reveals the influence of RPP and resting time on rheological curves. It can be seen from Figure 7 that the content of RPP and the resting time have a certain influence on the rheological properties of the cement paste.

Figure 7a shows that with the increase in RPP, the shear stress and its growth rate increase. Figure 7b shows that when the shear rate is lower than 50 s^−1^, the apparent viscosity of the pastes drops sharply with the increase in the shear rate and then rises slightly. After the pastes yielded, the apparent viscosity slightly increased with the increasing shear rate, reflecting that the pastes showed a slight shear-thickening behavior. With the increased content of RPP, the shear stress and apparent viscosity decrease. The extension of standing time does not change the rheological behavior of all pastes but increases the shear stress and apparent viscosity of each paste.

The rheological parameters of C-RPP pastes are shown in Table 7. From Table 7, the shear thickening degree drops when the content of RPP increases from 0.5% to 1.0%, but all rheological indexes for the C-RPP paste are larger than that in cement paste. With the increase in resting time, the rheological index goes up, illustrating a higher degree of shear thickening.

#### 3.2.2. Yield Stress and Plastic Viscosity

Figure 8 shows the yield stress and plastic viscosity of the C-RPP paste after standing for 15 min and 90 min.

Combined with the rheological parameters of C-RPP displayed in Table 7, it is clear that the $\tau_{0}$ of C-RPP paste decreases with the increase in RPP content. Furthermore, the longer the resting time, the greater the decline in the rate of yield stress. C-RPP pastes formed smaller $\tau_{0}$ than that in the C-LP pastes. After resting for 90 min, the increment of $\tau_{0}$ in C-RPP paste is less than that of the cement paste. Thus, the incorporation of RPP can effectively reduce the $\tau_{0}$ of the paste.

Figure 8 also describes the influence of the RPP content and the resting time on plastic viscosity in C-RPP paste. It can be seen from Table 7 and Figure 8 that the addition of RPP leads to the development of plastic viscosity. However, when the content of RPP increases from 0.5% to 1.0%, the plastic viscosity decreases slightly. It is visible that the plastic viscosity can be increased obviously by adding RPP. After resting for 90 min, the plastic viscosity of C-RPP paste does not increase significantly.

It is reported that [14] RPP dissolves in water to form a high viscosity emulsion, which can increase the viscosity after dispersing into the cement paste. During the process of blending with the cement mixture, the polymer particles of the RPP are continuously deposited, and the free water in the cement paste is adsorbed at the same time, which causes the volume fraction of the particles to increase. Additionally, RPP particles filled in the film and formed with the cement paste to develop a denser network structure, which can increase the plastic viscosity of the paste.

#### 3.2.3. Thixotropy

Figure 9 displays the thixotropic property of C-RPP paste with different RPP content and resting time. When 0.5% RPP was added to the cement paste, the thixotropic area of the paste decreased, but when the RPP content increased to 1.0%, the thixotropic area of the paste improved greatly. The thixotropic area of the C-R1.0 paste reaches from around 50 $\mathrm{Pa}\cdot s^{-1}$ to 70 $\mathrm{Pa}\cdot s^{-1}$ after the resting time extends from 15 min to 90 min. Therefore, an appropriate amount of flexible polymer RPP effectively improves the thixotropic property.

From the above results, it can be seen that the rheological parameters in the C-RPP paste varied with the content of RPP and resting time. When 1.0% RPP is added to the cement paste, the $\tau_{0}$ of the paste is reduced, and this increases the plastic viscosity appropriately. In addition, after resting for 90 min, the C-R1.0 paste still remains a dilatant, and its degree of shear thickening reduces and has an improved thixotropy, which is conducive to the formation of the internal structure of the paste and suits the characteristics of better working performance. The flexible polymer RPP has the functions of air entrainment and adsorption of free water and can be filled in the fresh cement paste to form a dense network structure [22]. As a result, the addition of RPP reduces the yield stress, improves the plastic viscosity and thixotropy, and it is also beneficial to improving the stability of the paste.

### 3.3. Rheology of C-HPMC Paste

#### 3.3.1. Rheological Curve

Figure 10 presents the influence of the HPMC content on the rheological curve of cement paste under different resting times. The incorporation of HPMC has a certain effect on the rheological curve, and it is more significant when the mixing amount reaches 0.075%.

From Figure 10a, it can be seen that when the shear rate is lower than 10 s^−1^, the shear stress of C-HPMC paste decreases sharply with the increase in shear rate, and the shear stress of the paste with more HPMC content greatly decreases. Then, with the increasing shear rate, the shear stress goes up, and the increased rate of shear stress in each paste is almost unchanged. As can be seen from Figure 10b, C-HPMC pastes take on shear-thinning after the pastes have yielded, indicating that the addition of HPMC maintains the rheological behavior of the cement paste. During the whole test, the apparent viscosity of the C-HPMC pastes goes down with the increase in shear rate.

The extension of the standing time does not change the rheological behavior of the pastes but increases the shear stress and apparent viscosity. However, after resting for 90 min, the greater the amount of the HPMC, the higher the shear stress and the lower degree of increase rate in apparent viscosity of the paste.

The rheological parameters of C-HPMC paste can be seen in Table 8. Comparing with rheological indexes of different HPMC incorporation under different resting time, it can be seen that the pastes mixed with 0.035% HPMC are dilatant; after standing for 90 min, the rheological index of pastes in the same group decreases to 1.082, illustrating that the degree of shear thickening decreases. However, pastes mixed with 0.075% HPMC are pseudoplastic. After standing for 90 min, the rheological index of pastes in the same group decreases to 0.697, illustrating that the degree of shear thinning increases.

#### 3.3.2. Yield Stress and Plastic Viscosity

Figure 11 displays the $\tau_{0}$ and plastic viscosity of C-HPMC paste under different resting times.

It is visible that a tiny amount of HPMC increases the yield stress significantly. Under the same resting time, the $\tau_{0}$ of the paste goes up markedly with the higher incorporation of HPMC. After resting for 15 min, $\tau_{0}$ increases from 56.396 Pa to 98.487 Pa when the content of HPMC increases from 0 to 0.035%. When the paste is mixed with 0.075% HPMC, the $\tau_{0}$ sharply increases to 178.783 Pa. Prolonging the resting time from 15 min to 90 min, the growth rate of yield stress in C-HPMC paste is lower than that of cement paste.

Figure 11 and Table 8 show the plastic viscosity and related parameters of C-HPMC paste. It is clear that after resting for 15 min, plastic viscosity in the paste with 0.035% HPMC increases slightly, and it increases rapidly from 1.500 $\mathrm{Pa}\cdot s^{n}$ to 2.561 $\mathrm{Pa}\cdot s^{n}$ when the content of HPMC increases to 0.075%. Therefore, the traces content of HPMC can lead to a prominent effect on increasing the plastic viscosity of the paste. Resting time has a minor impact on the plastic viscosity of C-HPMC pastes.

The network structure of the fiber molecules is interwoven with the network structure of the cement hydration products, as well as the initial hydration of the cement has an indirect effect on the increase in cellulose ether concentration, resulting in a significant increase in the viscosity of the cellulose ether-modified cement paste [15]. In addition, the HPMC molecules are able to produce internal solidification of free water through interactions between hydrogen molecules, reducing the availability of water in the first few hours and significantly increasing the viscosity around the mixed particles [23].

#### 3.3.3. Thixotropy

Figure 12 explains the thixotropic area of C-HPMC paste with different contents of HPMC and standing time. It can be seen from Figure 12 that the paste incorporated with 0.035% HPMC has a smaller thixotropic area than that of cement paste. When the content of HPMC increases to 0.075%, the thixotropic area increases. Additionally, the thixotropic area of each group goes down after prolonging the resting time. The addition of HPMC has a slight effect on the increasing thixotropic area when prolonging the resting time.

From the rheological parameters and thixotropic properties of C-HPMC paste, it is obvious that the incorporation of HPMC can maintain the rheological behavior of the cement paste. The addition of the HPMC increased the yield stress and plastic viscosity of the paste, and the increasing content of HPMC has a significant influence on raising the yield stress and plastic viscosity. However, the content of HPMC has little influence on thixotropy.

### 3.4. Rheology of C-L6-RPP Paste

#### 3.4.1. Rheological Curve

Figure 13 illustrates the rheological curves of the C-L6-RPP paste standing for 15 min and 90 min. The shear stress increases with the increasing shear rate, and the shear rate along with apparent viscosity also goes up with the increasing content of RPP. After resting for 15 min, the shear stress and apparent viscosity of paste mixed with 0.5% RPP are close to those of C-L6 pastes. After standing for 90 min, the shear stress and apparent viscosity of the C-R0.5 paste are slightly lower than that of the cement paste, indicating that 0.5% RPP has less influence on the C-L6 paste.

The addition of RPP in the C-L6 paste has the same rheological behavior. Each paste appeared to be shear thickening after the paste yielded at the shear rate of about 70 s^−1^. At the later stage, by increasing the resting time, the degree of shear thickening descends with the increase in RPP content.

Table 9 demonstrates the rheological parameters in the C-L6-RPP paste. It can be seen that with the increase in RPP content and resting time, the rheological index decreases. After resting for 90 min, all pastes shifted from dilatant to pseudoplastic; among all pastes, the paste which contained 1.0% RPP showed the greatest degree of shearing thinning.

#### 3.4.2. Yield Stress and Plastic Viscosity

Figure 14 displays the different $\tau_{0}$ and plastic viscosity under different RPP content and resting times for the C-L6-RPP paste. Rheological parameters are shown in Table 9.

It can be seen from Figure 14 that the incorporation of RPP has almost no effect on $\tau_{0}$ in the C-L6 paste after resting for 15 min. However, after resting for 90 min, $\tau_{0}$ achieves a certain increase in all pastes.

Since the incorporation of RPP increases the coverage of the polymer on the surface of cement and limestone particles, it creates a spatial barrier between the cement particles and the limestone powder, thereby weakening the van der Waals force between the particles. In conclusion, mixing a small amount of RPP could cause a slight drop in the yield stress of the paste [24].

Figure 14 also displays the plastic viscosity in the C-L6-RPP paste. As can be seen from Figure 14 and Table 9, C-L6 paste mixed with RPP slightly increases plastic viscosity. The extension of resting time leads to an inapparent influence on the plastic viscosity of the paste, which illustrates that the plastic viscosity of the C-L6-RPP paste is relatively stable and does not change significantly with the resting time. The incorporation of RPP into the C-LP paste effectively reduces the fluctuation in resting time, which effected the plastic viscosity of the paste and thus enhances its stability.

#### 3.4.3. Thixotropy

Figure 15 displays the changes in the thixotropic area on the C-L6 paste with different RPP contents and resting times. It can be seen from Figure 15 that the incorporation of RPP increases the thixotropy area compared with the cement paste, and with increasing standing time, the thixotropy area continued to increase, which is conducive for use in construction.

According to the results above, it can be concluded that the incorporation of RPP into the C-L6 paste does not change the rheological properties obviously. The incorporation of RPP has a low effect on the yield stress and plastic viscosity of the C-L6 paste but significantly improves the thixotropic stability of the paste after prolonging the resting time.

### 3.5. Rheology of C-L6-HPMC Paste

#### 3.5.1. Rheological Curve

Figure 16 displays the rheological curves of C-L6-HPMC paste under different resting times. From Figure 16, the increase in HPMC content and extended resting time has a remarkable effect on rheological curves.

The shear stress drops sharply at a low shear rate is less than 15 s^−1^ of the paste mixed with HPMC and then approaches a linear increase. The shear thinning appears in the process after the paste has yielded. In C-HPMC pastes, after resting for 15 min or 90 min, the greater the amount of HPMC, the higher the value of both shear stress and apparent viscosity. However, in C-L6-HPMC pastes, after standing for 15 min, the greater the amount of HPMC, the lower the shear stress and apparent viscosity.

Table 10 displays the rheological parameters in C-L6-HPMC paste. From the rheological index shown in Table 10, it can be seen that after the addition of HPMC, the rheological index of C-L6-HPMC paste almost takes on 1 in resting 15 min. However, after resting for 90 min, all pastes show shear thinning, especially the C-L6 paste mixed with 0.035% HPMC, and resting for 90 min shows the greatest degree of shearing thinning.

#### 3.5.2. Yield Stress and Plastic Viscosity

Figure 17 and Table 10 show the $\tau_{0}$, plastic viscosity, and rheological parameters of C-L6-HPMC paste under different contents of HPMC and resting times.

From Figure 17 and Table 10, it can be concluded that the traces amount of HPMC mixed with C-L6 paste can increase the $\tau_{0}$ visibly. After resting for 90 min, the $\tau_{0}$ of all paste increases by varying extents. The $\tau_{0}$ of the paste with only LP or HPMC shows a lower $\tau_{0}$. On the contrary, cement paste mixed with HPMC shows an upward tendency in $\tau_{0}$. However, the $\tau_{0}$ formed by C-L6-HPMC paste is much greater than that of both C-LP and C-HPMC pastes.

Figure 17 also shows the plastic viscosity in C-L6-HPMC paste influenced by HPMC and resting time. The plastic viscosity of C-L6-HPMC paste goes up with the increased content of HPMC. Moreover, the plastic viscosity growth rate of the paste mixed with HPMC is more significant than that with RPP. The plastic viscosity increases with the rising resting time. The influence of resting time on the C-L6-HPMC paste is greater than that of the C-L6-RPP paste.

Compared to the plastic viscosity of the C-L6 and C-HPMC pastes, the compounding of HPMC with LP into cement paste effectively enhances the workability of the paste, resulting in a more stable increment in the plastic viscosity with the increasing HPMC content.

#### 3.5.3. Thixotropy

Figure 18 shows the change in the thixotropic area for the C-L6-HPMC paste under different resting times. Figure 18 illustrates that by prolonging the resting time, the thixotropic area of C-L6-HPMC increased. Moreover, the higher the amount of HPMC, the greater the thixotropic property is, suggesting that the addition of HPMC in C-L6 paste has the benefit of improving the thixotropic property. Therefore, the organic thickening agent can not only effectively improve the yield stress and plastic viscosity of the paste but also the thixotropic property under the condition of a small amount of mixing.

From the above results, it can be seen that the addition of the HPMC into the C-L6 paste shows a greater influence on shear thinning. The incorporation of HPMC can significantly increase the yield stress and plastic viscosity and improve the thixotropic and working stability of the paste.

### 3.6. Rheology of C-L6-RPP-HPMC Paste

#### 3.6.1. Rheological Curve

Figure 19 shows the rheological curves of C-L6-RPP-HPMC paste. It can be seen that the paste contained 0.035% HPMC shows the highest shear stress. C-L6-RPP-HPMC paste shows the shear thinning in the whole shearing test. At a shear rate below 80 s^−1^, the apparent viscosity of the C-L6-R1.0-HPMC paste decreases rapidly but more slowly than the paste without RPP or HPMC. Moreover, the decline rate of apparent viscosity is much slower when the shear rate increases (greater than 80 s^−1^). In conclusion, the incorporation of HPMC and increasing resting time both cause a significant increase in shear stress and apparent viscosity of the C-L6-R1.0 paste.

Table 11 displays the rheological parameters of C-L6-RPP-HPMC paste. It can be seen in Table 11 that when resting time is prolonged from 15 min to 90 min that the pastes change from dilatant to pseudoplastic except for the C-L6-R1.0-H0.035 paste. The addition of HPMC makes the paste perform shear thinning and reduces the rheological index after resting for 90 min. In all, the resting time hardly influences the rheological indexes of C-L6-R1.0-H0.035 paste, which are close to 1.

#### 3.6.2. Yield Stress and Plastic Viscosity

Figure 20 describes the $\tau_{0}$ and plastic viscosity of C-L6-R1.0-HPMC paste under the condition of different HPMC content and resting time and with the value of $\tau_{0}$ shown in Table 11, it can be concluded that after resting for 15 min, the $\tau_{0}$ of the paste mixed with 0.035% HPMC is about 2.5 times that of the cement paste. With the increased standing time, the $\tau_{0}$ of the paste increases more obviously, which is about 60% higher than that of the paste resting for 15 min.

Compared with the $\tau_{0}$ of C-L6-R1.0 and C-L6-HPMC pastes, it can be clearly seen that the compounding of RPP and HPMC effectively improves the stability and workability of the paste. On the one hand, the compound blending of RPP and HPMC changes the decreasing tendency in the yield stress. On the other hand, the compounding of the two organic thickening agents has a synergistic effect on increasing the yield stress of the paste. That is, the $\tau_{0}$ of C-L6 paste mixed with only one organic thickening agent is much lower than that of the paste with the compounding of RPP and HPMC.

Figure 20 also represents the influence of the HPMC content and resting time on plastic viscosity in C-L6-R1.0-HPMC paste. With the value of plastic viscosity displayed in Table 11, it can be concluded that the incorporation of HPMC and the extended resting time are the factors that lead to an increase in the plastic viscosity of the C-L6-R1.0-HPMC paste.

Compared with the plastic viscosity of C-L6-R1.0 and C-L6-HPMC paste, the plastic viscosity in C-L6 paste is on the decline with the addition of RPP but increases with the addition of HPMC. Moreover, the compound effect of RPP and HPMC can better stabilize the plastic viscosity of the paste and also prevent the paste from showing excessive plastic viscosity after mixing with HPMC.

#### 3.6.3. Thixotropy

Figure 21 explains the thixotropic area in the C-L6-RPP-HPMC paste influenced by the HPMC content and resting time. As can be seen from Figure 21, the thixotropic property of the C-L6-RPP-HPMC paste increases due to the incorporation of HPMC. After resting for 90 min, the thixotropic area of the C-L6-R1.0-H0.035 paste increases by about 31.5%, and that of the cement paste increases by about 106%. According to the thixotropic property of all pastes mentioned above, the C-L6 paste mixed with RPP and HPMC shows the most stable thixotropic property with the increased content of the organic thickening agent, illustrating a better thixotropic property, as shown by the incorporation of RPP and HPMC adding with LP in cement paste.

The paste mixed with HPMC shows shear thinning during the whole rheological test. Moreover, the yield stress, plastic viscosity, and thixotropy of the paste are increased by adding the proper amount of HPMC. Furthermore, the effect of the standing time on the paste declines, and the stability of the paste improves.

### 3.7. Early-Age Hydration Products and Microstructure

#### 3.7.1. Early-Age Hydration Products

The TG and DTA curves of groups C, C-L6, C-R1.0, and C-H0.075 were tested so as to analyze the early-age hydration products affected by LP, RPP, and HPMC. Figure 22 shows the TG-DTA curves of each paste cured for three days. On the whole, TG and DTA curves show similar but slightly different trends, illustrating that LP, RPP, and HPMC have basically the same effect on the products generated by the hydration of the cement paste, but they all have a certain impact on the hydration process. From the peak values on the DTA curves, the hydration products content is different.

According to the corresponding temperature range of different peaks in the DTA curves in Figure 22a, the TG curves can be roughly divided into three stages: 80~150 °C is the first stage, and the endothermic peak is mainly produced by dehydration products such as C-S-H gel, ettringite, and AFm. The second stage is 400~550 °C, and the endothermic peak occurs at around 450 °C due to the mass loss caused by the removal of crystallized water from calcium hydroxide. The third stage is 650~800 °C, where the calcium carbonate minerals decompose at around 750 °C, resulting in a loss of carbon dioxide [23,25,26].

It can be obtained from the DTA curves that when the temperature reaches about 110 °C, there is only one strong endothermic peak appears in the cement paste, but within the first temperature stage of weight loss, two obvious weak endothermic peaks appear in C-L6 and C-R1.0 pastes while the strong endothermic peak in C-H0.075 paste is similar to that in cement paste. This is due to the fact that the heat absorption peaks between ettringite and C-S-H gel are indistinguishable because they overlap with each other when they dehydrate in cement paste, thus showing a stronger endothermic peak [27]. In the second stage, the endothermic peak of the C-L6 paste is the strongest, and it appears earlier than in other pastes. In the third stage, the endothermic peak of C-L6 paste is larger, which is because of the addition of LP, which contains more calcium carbonate minerals. When the temperature reaches about 700 °C, endothermic decomposition of calcium carbonate minerals occurs.

The temperature at which the weightlessness step appears in the TG curves corresponds to the DTA curves. From Figure 22b, it can be seen that the C-L6 paste is the first to show mass loss during the first weight loss step in each paste, indicating that the incorporation of LP has a facilitating effect on the hydration of the paste. In addition, because the C-L6 paste contains more calcium carbonate minerals, the third weight loss step is more obvious than the rest of the paste, suggesting C-L6 paste has the greatest mass loss in stage three. The paste mixed with an organic thickening agent (RPP, HPMC) has a lower weight loss compared to the cement paste, indicating that the incorporation of an organic thickening agent inhibited the generation of hydration products, which ultimately leads to a reduction in the degree of hydration. This is because the incorporation of a small amount of LP with a larger specific surface area disperses the cement particles and forms more nucleation sites to promote the hydration of the cement particles. On the contrary, the adsorption of the organic polymer particles into the cement particles and the film-forming effect prevent the particles from moving randomly of the paste and retards the hydration of the cement paste.

#### 3.7.2. Microstructure

Figure 23 displays SEM images of cement paste, C-L6 paste, C-R1.0 paste, C-H0.075 paste, and C-L6-R1.0-H0.035 paste cured for three days.

Figure 23a,b show the SEM image of cement paste. A large number of layered C-S-H hydration products have been formed after three days, as shown in Figure 23a. Further magnification at 10,000 times, as shown in Figure 23b, shows a great number of slender needle-like ettringite (Aft) crystals, which can be observed between the hydration products. C-S-H gel fills the gap and cavity between the solid particles, forming a denser structure.

Figure 23c,d present the SEM images of C-L6 paste with a magnification of 2000 times and 10,000 times, respectively, after curing for three days. The spindle shape of the C-S-H gel on the surface of the paste can be observed in the images, which shows a tighter connection and a denser microstructure of the C-L6 paste compared to the cement paste. The reason is that the cement paste is mixed with a small amount of LP with a larger specific surface area, which has a filling effect at the pre-hydration stage, providing more nucleation sites for the generation of hydration products and effectively refining the pore structure [28].

Figure 23e–h display the SEM images of C-R1.0 and C-H0.75 pastes cured for three days, respectively. From Figure 23e,g, the hydration products on the surface of the paste can be observed in a worm-like form. Furthermore, the strong air-entraining effect of RPP and HPMC can increase the air content of the cement paste, which results in a high porosity. Figure 23f,h are images magnified 10,000 times of C-R1.0 and C-H0.75 pastes, respectively; it can be observed that the surface of the modified cement hydration products is relatively smooth, with a small number of fine fluffy fibrous hydration products disperses outwards in clusters within the paste. It is because RPP and HPMC are fully dissolved in water and evenly dispersed in the mixture paste, which increases the viscosity of the paste and reduces the rate of ions binding to each other and the rate of water migration; thus, the formation of hydration products is delayed. In sum, the increasing rate of the aspect ratio of Aft is weak in the growth process, which shows as a short rod shape. Moreover, the growth of layered Ca(OH)_2_ is not mature, showing a fuzzy texture [29].

Figure 23i,j show the SEM images of the C-L6-R1.0-H0.035 paste. Since both RPP and HPMC can delay the hydration process of cement paste, it can be clearly observed in the images magnified by 10,000 times that RPP forms a polymer film structure in local areas of the cement paste and bridges between the hydration products, which in turn inhibits the growth of the hydration products such as Ca(OH)_2_, Aft, etc. Thus, the hydration products observed in the pictures, therefore, remain in an amorphous state.

It can be seen that the complementary effect of the three thickening agents slows down the hydration rate of cement paste, effectively improves the hydration of cement, and is more conducive to maintaining the stability of concrete in practical engineering. The existence of HPMC reduces the adsorption capacity of RPP on cement particles, which leads to more RPP particles remaining in particle gaps to form polymer films [30]. Additionally, the polymer film in Figure 23j has covered a certain area of the cement paste, and in other areas, the morphology of particle contour can still be observed in the polymer membrane, indicating that the film formation process in those areas stays in the stage of particle fusion [31].

## 4. Conclusions

At present, the combination of organic and inorganic thickening agents is rare in building materials, and the theoretical basis for the combination of organic and inorganic thickening agents is not mature enough. This paper provides a rheological basis for the composite use of organic and inorganic thickening agents through macroscopic and microscopic tests and provides a reliable basis for the replacement of inorganic thickening agents for cement in the future, and demonstrates the use of organic thickening agents to improve the stability and working performance of building materials.

This study investigated the rheological parameters, thixotropy, fluidity, early-age hydration products, and microstructure of cement paste containing tiny addition of limestone powder (LP), re-dispersible polymer powder (RPP), and hydroxypropyl methylcellulose ether (HPMC).

Combined with the relevant literature and the actual experimental conclusions obtained in this study, it can be concluded that the organic and inorganic thickening agents took on different thickening mechanisms. LP had a dense filling effect, while organic thickening agents had a film-forming and entanglement effect. The addition of LP, RPP, or HPMC into cement paste individually could increase the plastic viscosity; HPMC increased the yield stress but had little influence on the thixotropy. The addition of LP or RPP changed the rheological behavior and increased the fluidity of the cement paste, while HPMC had the opposite effect.

Compounding cement paste with LP, RPP, and HPMC greatly balanced the yield stress and plastic viscosity; it also improved thixotropy and reduced the impact of resting time on each rheological parameter. The C-L6-R1.0-H0.035 paste presented as a pseudoplastic, but the rheological indexes were close to 1 and hardly affected by the resting time.

In the TG and DTA curves, there were no great differences among the cement paste additions of RPP and HPMC. LP, RPP, and HPMC had a similar effect on hydration products. Both RPP and HPMC slowed down the hydration process, demonstrating that the C-L6-R1.0-H0.035 paste still stayed in the particle fusion stage when they were of three-days age.

## Figures and Tables

**Figure 1 materials-15-03159-f001:**
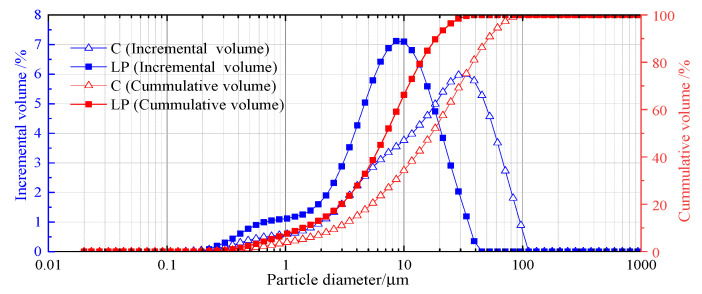
Particle size distribution of powders.

**Figure 2 materials-15-03159-f002:**
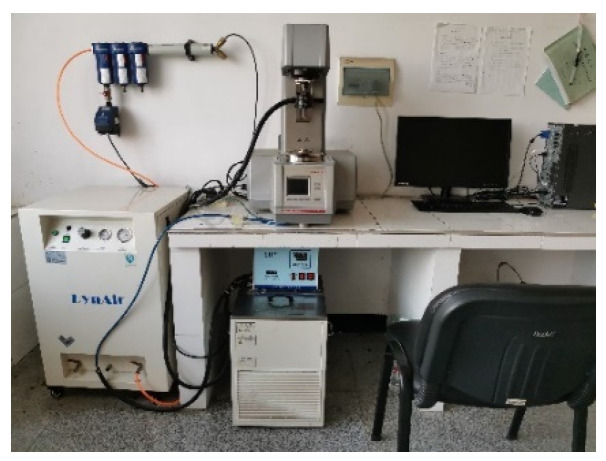
Anton Paar MCR 102 rheometer.

**Figure 3 materials-15-03159-f003:**
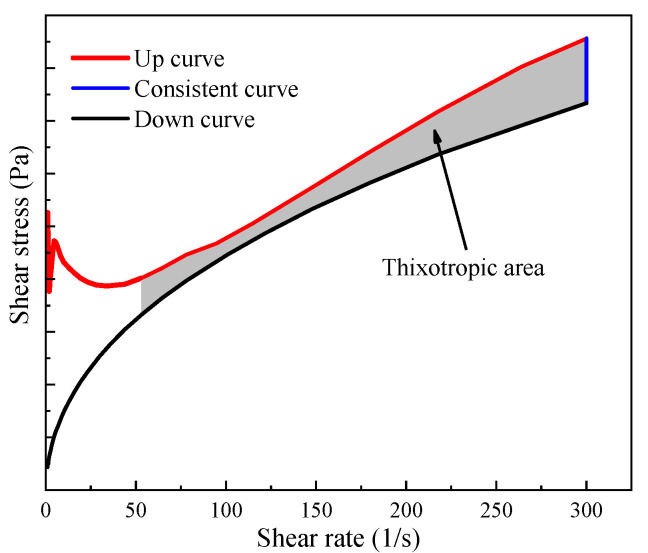
Thixotropic area of all pastes.

**Figure 4 materials-15-03159-f004:**
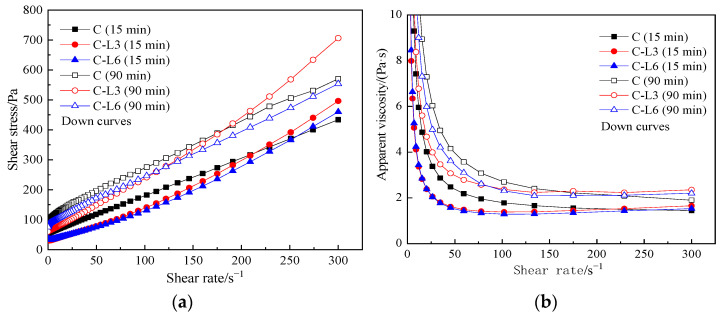
Rheological curves of C-LP paste. (**a**) shear stress vs. shear rate, (**b**) apparent viscosity vs. shear rate.

**Figure 5 materials-15-03159-f005:**
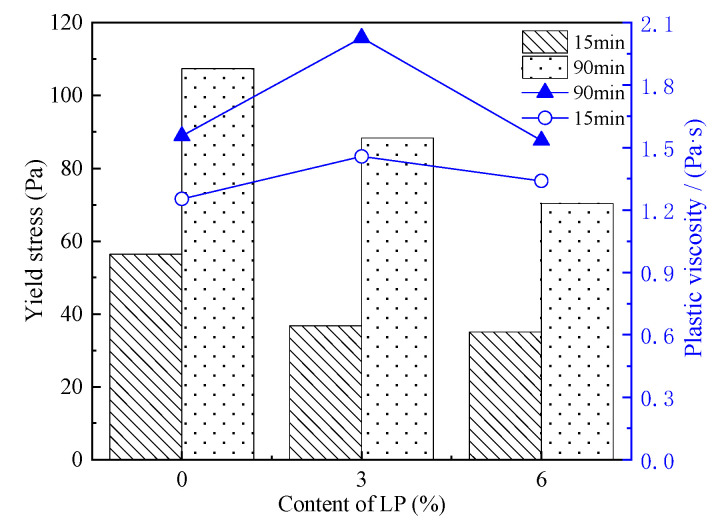
Yield stress and plastic viscosity of C-LP pastes.

**Figure 6 materials-15-03159-f006:**
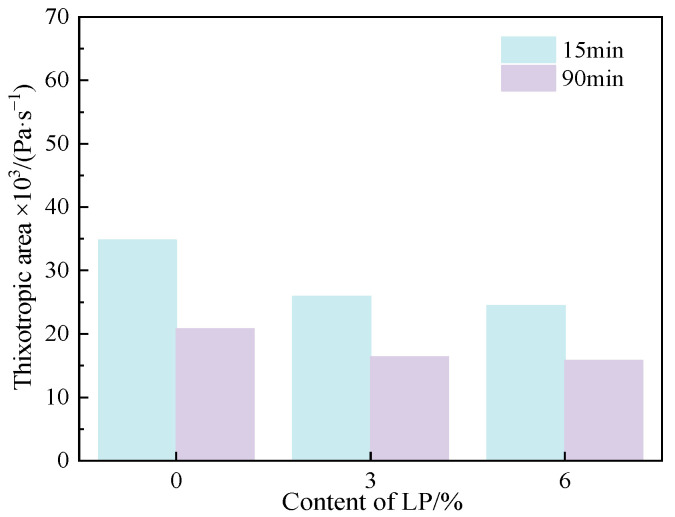
Thixotropic area of C-LP pastes.

**Figure 7 materials-15-03159-f007:**
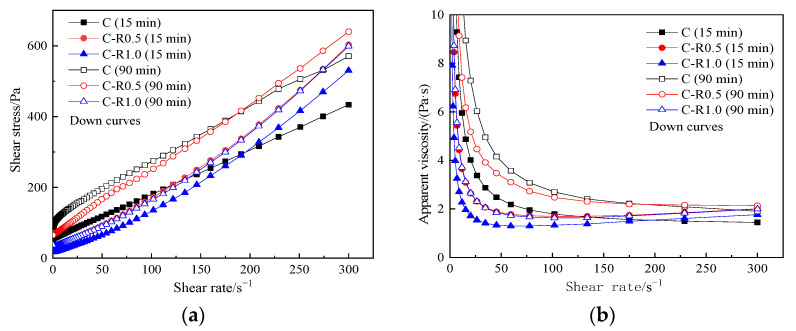
Rheological curves of C-RPP pastes. (**a**) shear stress vs. shear rate, (**b**) apparent viscosity vs. shear rate.

**Figure 8 materials-15-03159-f008:**
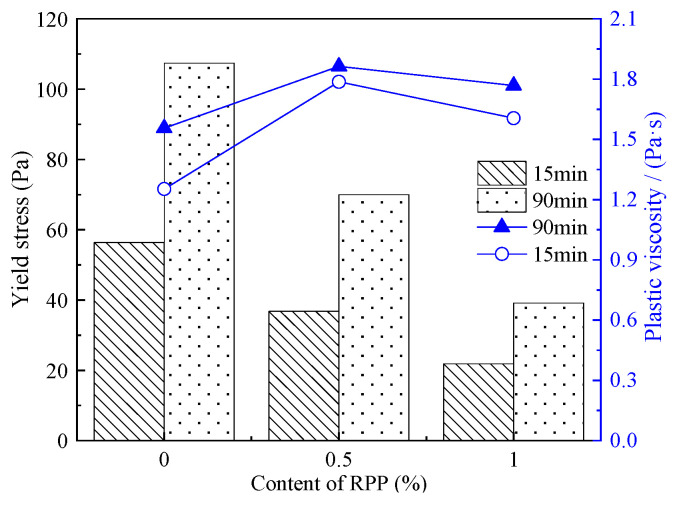
Yield stress and plastic viscosity of C-RPP pastes.

**Figure 9 materials-15-03159-f009:**
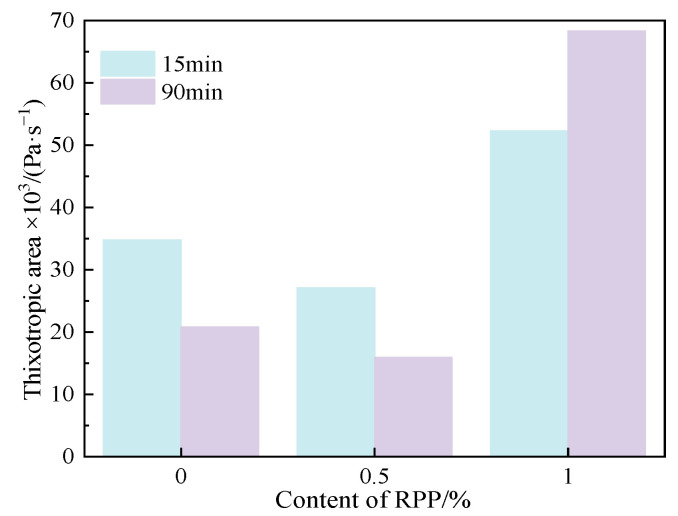
Thixotropic area of C-RPP pastes.

**Figure 10 materials-15-03159-f010:**
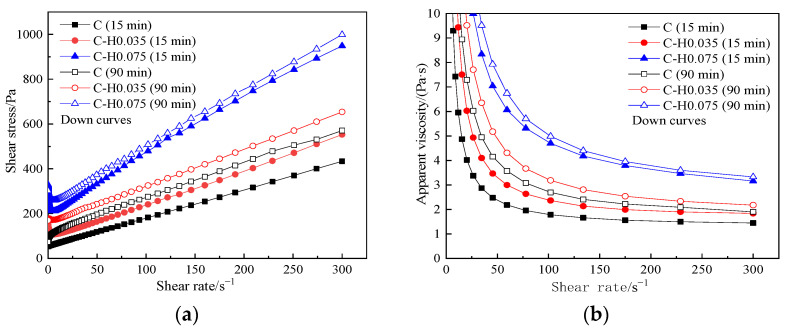
Rheological curves of C-HPMC pastes. (**a**) shear stress vs. shear rate, (**b**) apparent viscosity vs. shear rate.

**Figure 11 materials-15-03159-f011:**
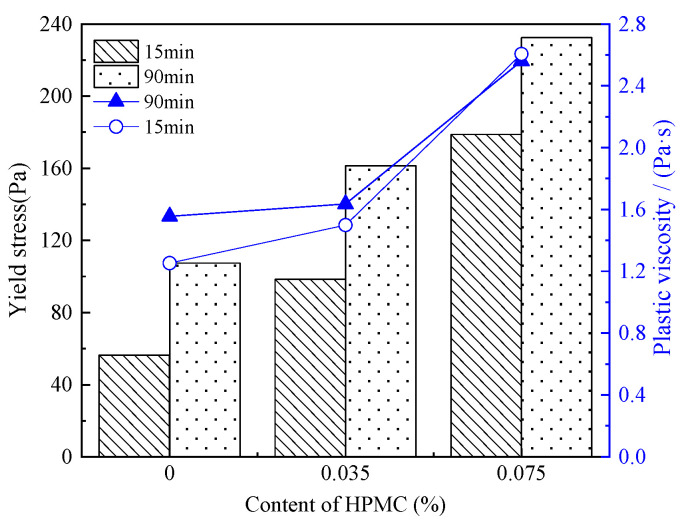
Yield stress and plastic viscosity of C-HPMC pastes.

**Figure 12 materials-15-03159-f012:**
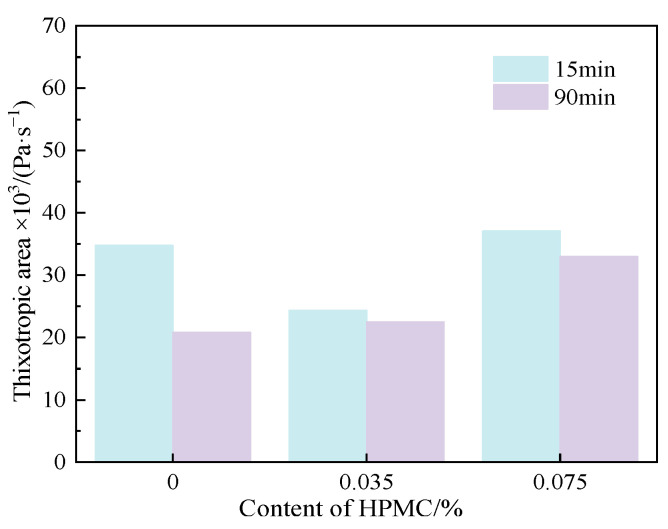
Thixotropic area of C-HPMC pastes.

**Figure 13 materials-15-03159-f013:**
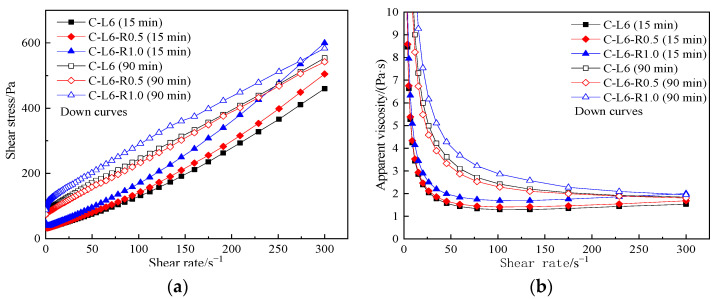
Rheological curves of C-L6-RPP pastes. (**a**) shear stress vs. shear rate, (**b**)apparent viscosity vs. shear rate.

**Figure 14 materials-15-03159-f014:**
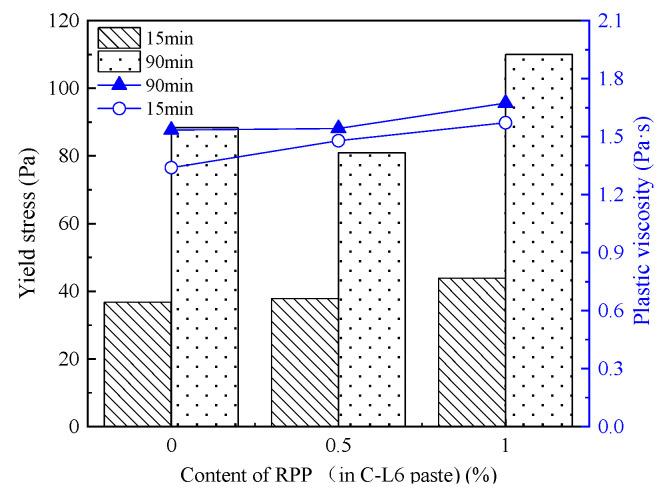
Yield stress and plastic viscosity of C-L6-RPP pastes.

**Figure 15 materials-15-03159-f015:**
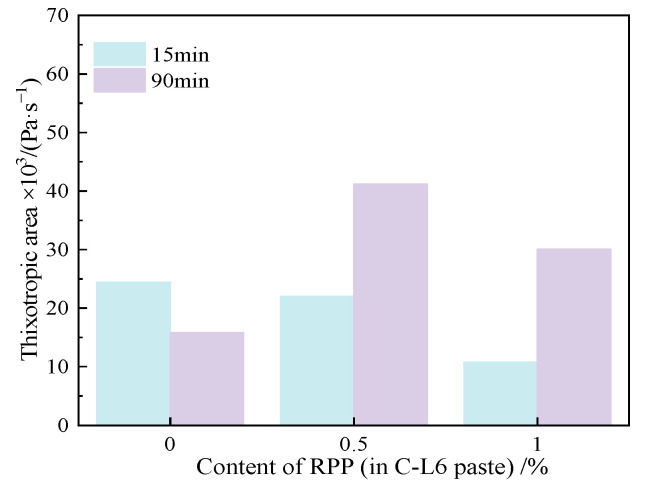
Thixotropic area of C-L6-RPP pastes.

**Figure 16 materials-15-03159-f016:**
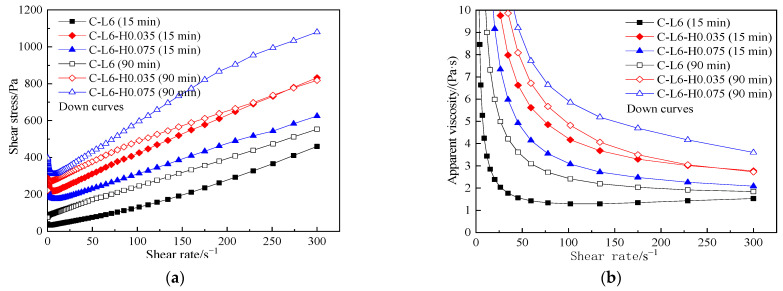
Rheological curves of C-L6-HPMC pastes. (**a**) shear stress vs. shear rate, (**b**) apparent viscosity vs. shear rate.

**Figure 17 materials-15-03159-f017:**
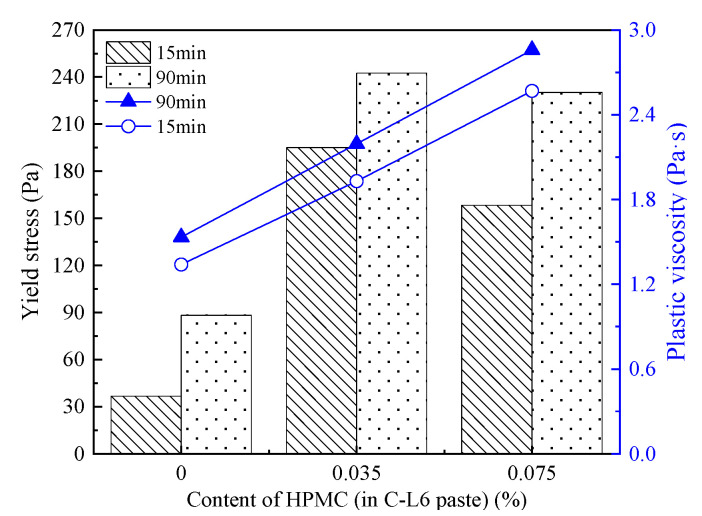
Yield stress and plastic viscosity of C-L6-HPMC pastes.

**Figure 18 materials-15-03159-f018:**
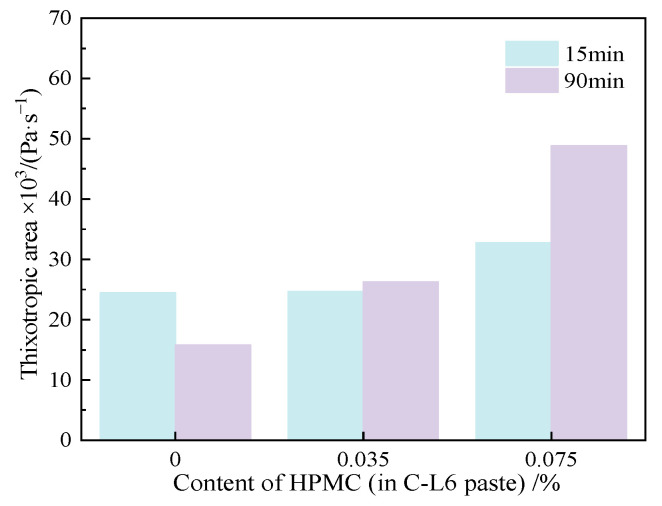
Thixotropic area of C-L6-HPMC pastes.

**Figure 19 materials-15-03159-f019:**
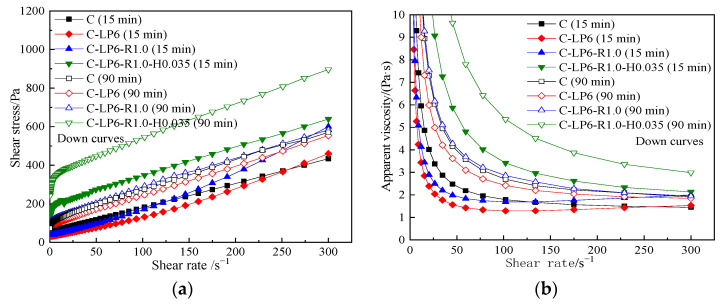
Rheological curves of C-L6-RPP-HPMC pastes. (**a**) shear stress vs. shear rate, (**b**) apparent viscosity vs. shear rate.

**Figure 20 materials-15-03159-f020:**
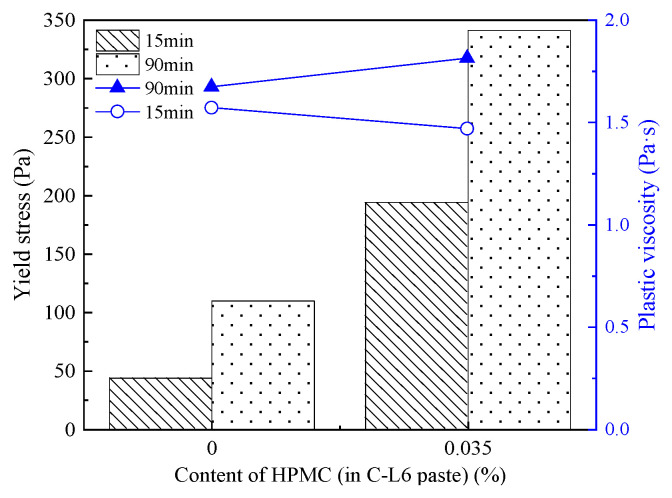
Yield stress and plastic viscosity of C-L6-R1.0-HPMC pastes.

**Figure 21 materials-15-03159-f021:**
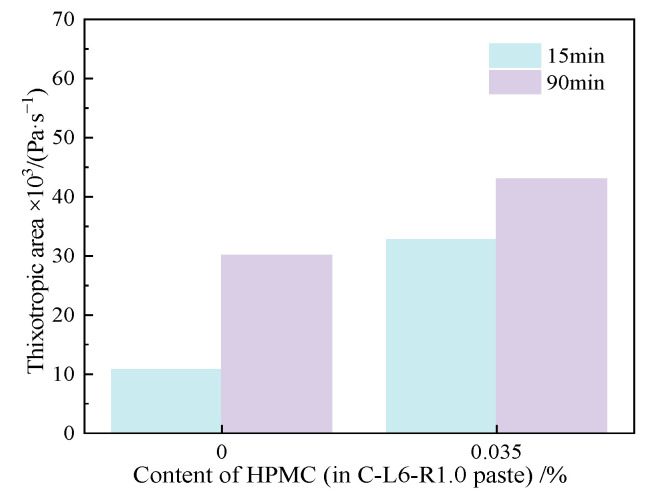
Thixotropic area of C-L6-RPP-R1.0-HPMC pastes.

**Figure 22 materials-15-03159-f022:**
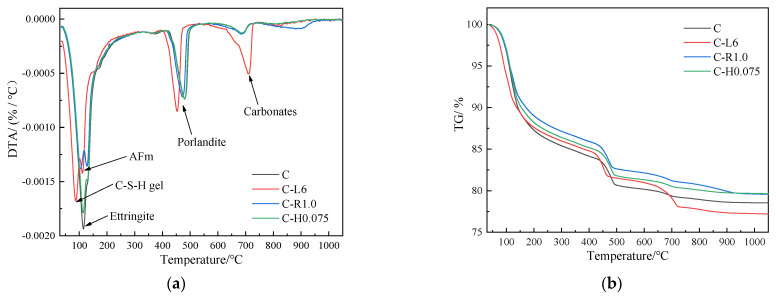
Thermogravimetric/differential thermogravimetry curves for 3 d of different pastes. (**a**) DTA curves, (**b**) TG curves.

**Figure 23 materials-15-03159-f023:**
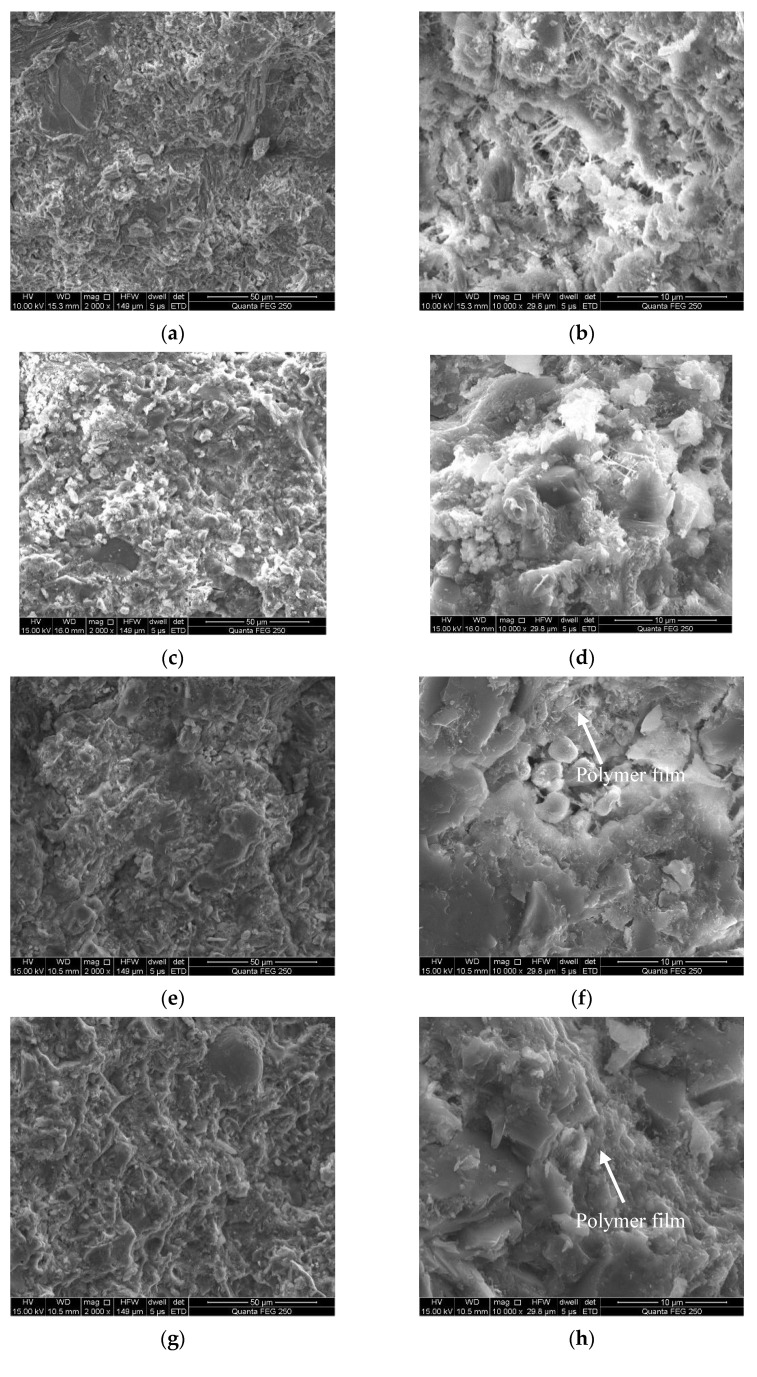
Microstructure of different pastes cured for 3 d. (**a**) Cement paste (Magnified 2000 times), (**b**) Cement paste (Magnified 10,000 times), (**c**) C-L6(Magnified 2000 times), (**d**) C-L6(Magnified 10,000 times), (**e**) C-R1.0(Magnified 2000 times), (**f**) C-R1.0 (Magnified 10,000 times), (**g**) C-H0.075 (Magnified 2000 times), (**h**) C-H0.075 (Magnified 10,000 times), (**i**) C-L6-R1.0-H0.035 (Magnified 2000 times), (**j**) C-L6-R1.0-H0.035 (Magnified 10,000 times).

**Table 1 materials-15-03159-t001:** Chemical composition of cement (w/%).

	SiO_2_	Al_2_O_3_	Fe_2_O_3_	CaO	MgO	SO_3_
PC	20.95	5.19	3.83	64.29	1.85	2.95

**Table 2 materials-15-03159-t002:** Mineral composition of cement (w/%).

	C3S	C2S	C3A	C4AF
PC	59.38	15.59	7.28	11.64

**Table 3 materials-15-03159-t003:** Physical properties of cement.

Fineness 0.08/%	Specific Surface Area/(m^2^/kg)	Density /(g/cm^3^)	Standard Consistency/%	Setting Time/min		Flexural Strength/MPa	Compressive Strength/MPa
				Initial	Final	3d	3d
0.5	346	3.12	24.60	96	158	6.5	28.0

**Table 4 materials-15-03159-t004:** Medium particle size and specific surface area of cement and limestone powder.

Material	$\mathrm{The}\;\mathrm{Particle}\;\mathrm{Size}\;\left(\mu m\right)$	Specific Surface Area/(m^2^·kg^−1^)
C	17.194	345
LP	7.077	1104

**Table 5 materials-15-03159-t005:** Mix proportion of cement compound paste.

NO.	Sample	Mix Proportion/g					
		C	LP	RPP	HPMC	SP	W
1	C	500	0	0	0	0.85	160
2	C-L3	485	15	0	0		
3	C-L6	470	30	0	0		
4	C-R0.5	500	0	2.5	0		
5	C-R1.0	500	0	5.0	0		
6	C-H0.035	500	0	0	0.175		
7	C-H0.075	500	0	0	0.375		
8	C-L6-R0.5	470	30	2.5	0		
9	C-L6-R1.0	470	30	5	0		
10	C-L6-H0.035	470	30	0	0.175		
11	C-L6-H0.075	470	30	0	0.375		
12	C-L6-R1.0-H0.035	470	30	5	0.175		

**Table 6 materials-15-03159-t006:** Rheological parameters of C-LP pastes.

NO.	Resting Time (min)	$\tau_{0}\;\left(\mathrm{Pa}\right)$	$K\;\left(\mathrm{Pa}\cdot s^{n}\right)$	n	R
C	15 min	56.4	1.253	1.001	0.9999
	90 min	107.4	1.555	0.794	0.9951
C-L3	15 min	36.8	1.057	1.323	0.9995
	90 min	88.3	1.154	1.154	0.9987
C-L6	15 min	35.1	1.340	1.353	0.9993
	90 min	70.3	1.533	0.932	0.9989

**Table 7 materials-15-03159-t007:** Rheological parameters of C-RPP pastes.

NO.	Resting Time (min)	$\tau_{0}\;\left(\mathrm{Pa}\right)$	$K\;\left(\mathrm{Pa}\cdot s^{n}\right)$	n	R
C	15 min	56.4	1.253	1.001	0.9999
	90 min	107.4	1.555	0.794	0.9951
C-R0.5	15 min	36.8	1.787	1.293	0.9994
	90 min	70.0	1.863	0.998	0.9992
C-R1.0	15 min	21.8	1.605	1.361	0.9998
	90 min	39.1	1.768	1.336	0.9994

**Table 8 materials-15-03159-t008:** Rheological parameters of C-HPMC pastes.

NO.	Resting Time (min)	$\tau_{0}\;\left(\mathrm{Pa}\right)$	$K\;\left(\mathrm{Pa}\cdot s^{n}\right)$	n	R
C	15 min	56.4	1.253	1.001	0.9999
	90 min	107.4	1.555	0.794	0.9951
C-H0.035	15 min	98.5	1.500	1.303	0.9755
	90 min	161.5	1.636	1.082	0.9988
C- H0.075	15 min	178.8	2.607	0.739	0.9455
	90 min	232.5	2.561	0.697	0.9873

**Table 9 materials-15-03159-t009:** Rheological parameters of C-L6-RPP pastes.

NO.	Resting Time (min)	$\tau_{0}\;\left(\mathrm{Pa}\right)$	$K\;\left(\mathrm{Pa}\cdot s^{n}\right)$	n	R
C-L6	15 min	36.8	1.340	1.353	0.9993
	90 min	88.3	1.533	0.932	0.9989
C-L6-R0.5	15 min	37.8	1.478	1.347	0.9992
	90 min	81.0	1.541	0.985	0.9996
C-L6-R1.0	15 min	43.9	1.572	1.339	0.9993
	90 min	110.0	1.673	0.825	0.9977

**Table 10 materials-15-03159-t010:** Rheological parameters of C-L6-HPMC pastes.

NO.	Resting Time (min)	$\tau_{0}\;\left(\mathrm{Pa}\right)$	$K\;\left(\mathrm{Pa}\cdot s^{n}\right)$	n	R
C-L6	15 min	36.8	1.340	1.353	0.9993
	90 min	88.3	1.533	0.932	0.9989
C-L6-H0.035	15 min	195.0	1.930	0.944	0.9698
	90 min	242.5	3.593	0.797	0.9963
C-L6-H0.075	15 min	158.3	2.568	1.027	0.9466
	90 min	230.2	2.858	0.862	0.9633

**Table 11 materials-15-03159-t011:** Rheological parameters of C-L6-RPP-HPMC pastes.

NO.	Resting Time (min)	$\tau_{0}\;\left(\mathrm{Pa}\right)$	$K\;\left(\mathrm{Pa}\cdot s^{n}\right)$	n	R
C	15 min	56.4	1.253	1.001	0.9999
	90 min	107.4	1.555	0.794	0.9951
C-L6	15 min	36.8	1.340	1.353	0.9993
	90 min	88.3	1.533	0.932	0.9989
C-L6-R1.0	15 min	43.9	1.572	1.339	0.9993
	90 min	110.0	1.673	0.825	0.9977
C-L6-R1.0-H0.035	15 min	194.2	1.469	0.998	0.9952
	90 min	341.3	1.813	0.985	0.9859

## Data Availability

The data presented in this study are available on request from the corresponding author.

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
