# Peer review of "Rheological Properties and Early-Age Microstructure of Cement Pastes with Limestone Powder, Redispersible Polymer Powder and Cellulose Ether"

_materials, 2022, doi:10.3390/ma15093159_

Round 1
Reviewer 1 Report
The manuscript is about the investigation of rheological properties, morphology, and DSC/TGA of cement pastes with some additions. Unfortunately, the rheological properties are presented in the linear coordinates, so they cannot be fully evaluated. It is necessary to make the appropriate edits before the next full review.
Line 108 How the particles’ size distributions were obtained? It should be noted in Methods.
Lines 175-509 Fig 3, etc. Flow curves must be given in log-log coordinates. It is in logarithmic coordinates that one can understand and trace the behavior of complex fluids under flow or oscillatory influences. Because the change in magnitude spans many orders of magnitude. The corresponding charts need to be rearranged.
The review of rheological properties will be possible only after bringing the graphs to the log coordinates
Line 595 Judging by the presented SEM images, what morphology type is the most preferable?
Minor flaws:
Line 26 “SEM images” is more correct.
Line 92 etc. Space symbol between a number and a designation should be. Values must be rounded correctly.
Author Response
Thank you for your comments on this article! Please see tha attachment.

Reviewer 2 Report
Title: Rheological Properties and Early-age Microstructure of Cement Pastes…
Manuscript ID: materials-1688756
Authors: Feng et al.
Dear Authors,
Thank you for the opportunity to read your article. I found the topic is interesting and fundamental. Generally speaking, there are some results presented in order to capture some trends, but the introduction and methods need more clear explanation while the results need discussion with fair point of view. I suggest that this article will be revised extensively before its re-submission for another review process if applicable. As a conclusion, I recommend its major revision at this state.
I hope my comments are helpful.
Good luck,
A reviewer
Major concerns:
“Keywords”
-Please consider listing keywords that are not used in the article title.
“1. Introduction”
-Line 64: “…H-B model…”->Please consider providing the full spelling of H-B model and its reference. The Herschel-Bulkley model?
-Lines 79-85: “…LP, HPMC, RPP…”->Please consider stating why you selected those agents among others. Also, please consider dividing this statement into two since it is too long.
“2. Materials and Methods”
“2.1. Raw materials and admixture proportion”
-Fig.1 and Table 4: Please consider describing and discussing more about them.
-Fig.1: Please indicate what the data plotted with (1) red filled square and (2) red blank triangle correspond.
“2.2. Testing methods”
“2.2.1. Rheological test and analytical model”
-Lines 149-152: In this section, please consider providing relevant reference(s) indicating the correlation between n and shear thickening/thinning.
“2.2.2. Fluidity test”
-In this section, please consider stating how you analyzed the obtained data.
“2.2.4. SEM analysis”
-In this section, please consider providing more details about the measurements. For example, please consider providing more detail information about your SEM imaging, including the way you deposited your sample(s) in an SEM chamber, detector type (SE? BSE?), accelerating voltage, working distance. Also, please consider explaining how you analyzed your images. Those information would be helpful for future researchers. This comment also applied to all the characterization methods introduced in this section.
doi.org/10.1016/j.actamat.2005.12.014
doi:10.3390/electronics8101202
“3. Results and discussion”
-In general, please consider discussing your results. Detailed comments can be found below.
“3.1. Rheology of C-LP paste”
“3.1.1. Rheological curve”
-Lines 184-186: “In the first half of the test, the shear stress of blank cement past is higher…”->Please be more specific and citing numbers, and also consider discussing the results. In other words, please discuss why the first half and second half of shear rate provides different behavior for the blank cement past and C-LP pastes.
-Fig.5 (and elsewhere): Please clearly state your samples and conditions in the figure title.
-Lines 202-204: “After prolonging the resting time…dilatant…pseudoplastic…”->Please consider discussing the meaning of this result in scientific and engineering point of view.
“3.1.2. Yield stress and plastic viscosity”
-Lines 214-215: “…inter-particle interaction force…the repulsive force…”->Please consider naming and explaining the corresponding forces together with relevant reference(s).
-Lines 217-218: “…from 346 m2 kg-1 to 349.5 m2 kg-1. Thus, the τ0 in C-LP pastes decreased with the increasing content of LP.”->(a) Please consider explaining your experiments to obtain the specific surface area. (b) Please consider providing other potential reasons to explain the decrease in τ0 since the decrease in the specific surface area is about 1% while the decrease in more than 30%. Also the decrease in the specific surface area is more than 10% in literature [19].
-Figs. 6 and 7 (and elsewhere): Please consider adding error bars.
-Line 220-221: “…increases the plastic viscosity of the pastes.”->Please consider revising this statement since not all your experimental results agree with your statement. As seen in Fig.6, only the conditions the plastic viscosity increases are: from 0 to 3% LP at 15 and 90 min resting time. Also, for 15 min, the change is very small and can be within the error range.
-Sections 3.1 to 3.6: Please consider discussing the difference among different formulations. In the current article at least till the section 3.6, there is no discussion among different formulation using your rheology results.
“3.7.2. Microstructure”
-Please consider correlating your SEM images and interpretation with your rheology results.
“4. Conclusions”
-You may state future perspectives in Conclusions.
Minor concerns:
-Please consider polishing English more. You may use some of my comments above and below for this purpose.
-Line 36: “In practice engineering…”-> In practical engineering…
Author Response
Thank you for your comments on this article! Please see the attachment.

Reviewer 3 Report
Dear Feng et al.,
The manuscript “Rheological Properties and Early-age Microstructure of Cement Pastes with Limestone Powder, Redispersible Polymer Powder and Cellulose Ether” (materials-1688756) by Feng et al. investigate the synergistic effects of organic and inorganic thickening agent on rhe-9 ological properties of cement paste, the rheological parameters, thixotropy of cement paste containing limestone powder (LP), redispersible polymer powder (RPP) and hydroxypropyl methyl cellulose ether (HPMC) by the Anton Paar MCR rheometer at different resting time The topic is interesting, but I think this article should reconsider after proper changes in major revision for publication in Materials. Some of my specific comments are below:
- In the abstract section (line 9-27), the authors should add quantitative results rather than only qualitative results.
- Describe the novelty of the article made by the author? From the results of my evaluation, it seems that many similar published works adequately explain what you have raised in the current manuscript. If there are something others really new in this manuscript, please highlight it more clearly in the introduction section (line 30-85).
- The state of the art and the significance of the current study are not clearly present, the authors should highlight it more advanced in the introduction section (line 30-85).
- In the introduction section (line 30-85), the authors should explain the previous research conducted and its shortcomings. It will uphold the research gap that you filled with your research novelty. I recommend the authors elaborate on their introduction section. Do not forget to attention carefully my previous comments on numbers 2 and 3.
- Since this manuscript study of mechanical performance of cement, I would encourage and advise the authors to adopt some of the specific additional references related to medical implants, cement as fixation would enhance its overall ability published by MDPI in the introduction section (line 30-85) as follow:
-
- Tresca Stress Simulation of Metal-on-Metal Total Hip Arthroplasty during Normal Walking Activity. Materials (Basel). 2021, 14, 7554. https://doi.org/10.3390/ma14247554
- In the materials and methods section (line 86-173), the authors should add one systematic figure to illustrate the workflow of experimental testing in the present study to make the reader more interested and easier to understand rather than only using dominant text to explain.
- The author must provide a detailed specification and use condition more detail regarding all tools used in the research carried out so that the reader can estimate the accuracy and differences in the results that the authors describe due to the use of different tools in future studies.
- In the Results and discussion section (line 174-595), the authors are advised to compare the results they obtain with previous similar/identical studies if it is possible.
- In the last paragraph before conclusion section (after line 595), the authors should add of one paragraph about the limitations of the presented review.
- The conclusion (line 596-619) of the present manuscript is not solid. Further elaboration is needed. Also, make it intho paragraph, not point-by-point as in present form.
- Further research needs to be explained in the conclusion section (line 596-619).
- In the whole of the manuscript, the authors sometimes made a paragraph only consisting of one or two sentences that made the explanation not clearly understood. The authors need to extend their explanation to become a more comprehensive paragraph. In one paragraph, it is recommended to consist of at least 3 sentences with 1 sentence as the main sentence and the other sentences as supporting sentences.
- I see some errors on English in some areas of the present manuscript. To improve the quality of English used in this manuscript and make sure English language, grammar, punctuation, spelling, and overall style are correct, further proofreading is needed. As an alternative, the authors can use the MDPI English proofreading service for this issue.
- Please make sure the authors have used the Materials, MDPI format correctly. The authors can download published manuscripts by Materials, MDPI, and compare them with the present author's manuscript to ensure typesetting is appropriate. For example:
-
- The information of Funding, Institutional Review Board Statement, Informed Consent Statement, and Data Availability Statement are not provided
- References typesetting is error
- And other
I am pleased to have been able to review the author's present manuscript. Hopefully, the author can revise the current manuscript as well as possible so that it becomes even better. Good luck for the author's work and effort.
Best regards,
The Reviewer
Author Response

(The authors gave the same response as above.)

Round 2
Reviewer 1 Report
Thanks to the authors for the response. I got answers to all the main questions, a couple of small remarks:
The tables taken for comparison have too many digitals. The tau0 data in Table 6-11 is redundant and reported accuracy is questionable. For example, it is not correct to give the value 56.396. It should be 56 or 56.4,
The same is true for the other experimental parameters.
Line 128 In accordance with the MDPI rules, the device model, country, city must be indicated.
Author Response
Dear Reviewer 1,
Thanks for providing us with this great opportunity to submit a revised version of our manuscript. We appreciate the detailed and constructive comments provided by the reviewers. We have carefully revised the manuscript by incorporating all the suggestions by the review panel. Please see the attachment for detailed modification.
We hope this revised manuscript has addressed your concerns, and look forward to hearing from you.
Sincerely,
The authors

Reviewer 2 Report
Dear Authors,
As all the comments and concerns were addressed, I suggest that the journal accepts the manuscript for its publication.
Best regards,
A reviewer
Author Response
Dear Reviewer,
Thank you for your letter and the comments concerning our manuscript. Those comments are all valuable and very helpful for revising and improving our paper, as well as the important guiding significance to our researches. Thanks very much for your kind work and consideration on publication of our paper.
Sincerely,
The Authors
Reviewer 3 Report
Dear Fenget al.,
After carefully reading the author's revised manuscript entitled "Rheological Properties and Early-age Microstructure of Cement Pastes with Limestone Powder, Redispersible Polymer Powder and Cellulose Ether" (materials-1688756) by Feng et al., The authors have been made significant improvements in the revised manuscript. Also, all of the issue in my review report has been addressed precisely.
With my pleasure, I recommend the manuscript should be accepted for publication on Materials.
Best regards,
The Reviewer
Author Response

(The authors gave the same response as above.)
